# Synthesis, Photoswitching Behavior and Nonlinear Optical Properties of Substituted Tribenzo[*a*,*d*,*g*]coronene

**DOI:** 10.3390/molecules28031419

**Published:** 2023-02-02

**Authors:** Xueqing Li, Jie Zhao, Wei Wang, Yiming Li, Yunfei Li, Shuyun Zhou, Jinchong Xiao

**Affiliations:** 1Key Laboratory of Chemical Biology of Hebei Province, College of Chemistry and Environmental Science, Hebei University, Baoding 071002, China; 2Key Laboratory of Photochemical Conversion and Optoelectronic Materials, Technical Institute of Physics and Chemistry, Chinese Academy of Sciences, Beijing 100190, China

**Keywords:** tribenzocoronene, crystal structure, physical property, nonlinear optical property

## Abstract

A family of tribenzocoronene derivatives bearing various substituents (**3**) were constructed through the Diels–Alder reaction, followed by the Scholl oxidation, where the molecular structure of **3b** was determined via single crystal X-ray diffraction analysis. The effect of substitution on the optical and electrochemical property was systematically investigated, with the assistance of theoretical calculations. Moreover, the thin films of the resulting molecules **3b** and **3e** complexed with fullerene produced strong photocurrent response upon irradiation of white light. In addition, **3b** and **3e** exhibit a positive nonlinear optical response resulting from the two-photon absorption and excited state absorption processes.

## 1. Introduction

The construction of structurally defined polycyclic aromatic hydrocarbons (PAHs) has attracted substantial interest during the past several decades because such molecules can be usually seen as a segmental model for defects of graphene possessing interesting physical properties and can be used in organic electronics including laser, photodetectors, organic light emitting diodes, organic field effect transistors and organic solar cells [1,2,3,4]. Among them, curved π-conjugated derivatives provide us with more room to deepen our understanding of the anomalous hexagon arrays, which can be obtained through the implementation of armchair, cove, fjord regions, and the embedment of four-, five- seven- and eight-membered rings [5,6,7,8]. Undoubtedly, the edge and size can affect the optoelectronic and magnetic properties to a great extent, leading to different aromaticity, energy levels and band gaps. Meanwhile, if the heteroatoms or heterorings were doped into the parent frameworks, the resulting heteroarenes exhibit some appealing behaviors such as ease of synthesis, tailoring physical property and molecular stability [9,10,11,12]. More interestingly, the introduction of some functional groups including electron-withdrawing groups and electron-donating groups into the π-systems is a straightforward method for selective modification and functionalization. As a highly symmetric (D_6h_) molecule, coronene is the subject of considerable investigations owing to its tailoring optoelectronic properties. The self-assembly of a single coronene can form regular nanowires used for optoelectronic devices [13]. More strikingly, this “standard” six-membered ring-fused molecule cocrystallizes with different acceptors, including 7,7,8,8-tetracyanoquinodimethane (TCNQ), 2,3,5,6-tetrafluoro-7,7,8,8-tetracyanoquinodimethane (F4TCNQ), 1,2,4,5-tetracyanobenzene (TCNB), napthalenetetracarboxylic diimide via a weak interaction [14,15,16,17,18]. In addition, the functionalization of appropriate precursors can generate expanded coronene derivatives. All the observations stimulate us to prepare novel arenes bearing a coronene unit and to investigate the physical properties.

The wide application of laser in optic devices provides great convenience in military and civil aspects [19,20,21]. However, the big injury risk of laser asks researchers to acquire excellent optical limiting materials to decrease the laser intensity in order to protect the eyes and optical devices. At present, organic materials such as fullerene derivatives, phthalocyanine, graphene and metallated graphdiyne can compete with inorganic counterparts over flexibility, machinability and quick response [22,23,24,25]. In comparison to gapless graphene, coronene and its derivatives are well defined, as well as possessing a controllable physical property and a high charge-carrier mobility. Hirata et al. found that β-estradiol doped with deuterated coronene could present large reverse saturable absorption characteristics with sunlight power level [26]. More recently, organic co-crystals based on coronene and naphthalenediimide have exhibited an enhanced nonlinear optical response and charge transfer with the increase in the intermolecular interaction in the group of Wang [27]. Until now, the optical limiting property of such functionalized coronene derivatives has been limited.

In this work, we strategically synthesized a family of substituted tribenzo[*a*,*d*,*g*]coronene derivatives (**3a**–**3e**, Figure 1). The molecular structure of 5,10-di-*tert*-butyl-15-chlorotribenzo[*a*,*d*,*g*]coronene (**3b**) is determined through single crystal X-ray diffraction. All of them emit green fluorescence. The complexes of **3b**/**3e**-C_60_ produce a strong photoresponse under irradiation of white light. Moreover, **3b** and **3e** exhibit nonlinear optical performance and the possible mechanism is caused by the two-photon absorption and excited state absorption processes. Clearly, systematic studies may be instructive to design and approach coronene-containing derivatives.

## 2. Results and Discussion

The synthesis of substituted tribenzo[*a*,*d*,*g*]coronenes is depicted in Figure 1. The key intermediate **2** was substituted with 2,7-di-*tert*-butyl-9,14-diphenyldibenzo[*de*,*qr*]tetracenes, which was achieved in medium yield via the classical Diels–Alder reaction between black solid 2,7-di-*tert*-butyl-9,11-diphenyl-10H-cyclopenta[*e*]pyren-10-one (**1**) and substituted 2-aminobenoic acid under the existence of isopentyl nitrate in degassed 1,2-dichlroethane. It should be mentioned that molecules **3a** and **3c** were prepared according to the synthetic route [28]. A Scholl reaction of **2** in anhydrous dichloromethane with the assistance of triflic acid (TfOH) and 2,3-dichloro-5,6-dicyano-p-benzoquinone (DDQ) provided **3a**–**3d**. Treatment of **3c** with CuCN in 1-methyl-2-pyrrolidinone (NMP) generated 5,10-di-*tert*-butyltribenzo[*a*,*d*,*g*]coronene-15-carbonitrile (**3e**) in an isolated 24% yield. All the new compounds were purified by silica gel column chromatography and characterized through ^1^H NMR, ^13^C NMR and HR-MS (Appendix A). More importantly, such resultant derivatives bearing various substituents should provide more room for their selective modification and functionalization.

To further prove the molecular structures and examine the arrangement in the solid state, flake-like single crystals of **3b** were obtained by slow evaporation of 1,2-dichloroethane (DCE) and acetonitrile solution. It should be noted that no crystals suitable for single crystal X-ray analysis of **3a**, **3c**, **3d** and **3e** were formed under a similar operation condition. Molecule **3b** adopts monoclinic space group C2/c with Z = 8. The unit cell dimensions are *a* = 26.684(3) Å*, b* = 13.5714(13) Å, *c* = 19.003(2) Å, β = 100.02(4)^o^ (Appendix A). As can be seen from Figure 1a, all the benzene rings are not in one plane, which is different from the parent coronene unit [29]. More interestingly, the benzo moieties on the pyrene and the terminal chlorobenzene unit in the horizontal tetracene part bend to the same side, and thus **3b** can form a reclining-chair configuration (Figure 1b). Similar architectures were observed in the twistarenes observed in our group [30]. Molecule **3b** can stack in column style, where the distance between the naphthalene in the pyrene unit is 3.63 Å (Figure 1c), which implies that π-π stacking interaction is absent [31].

The optical properties were manifested via UV-visible absorption and fluorescence spectra in a solution. As shown in Figure 2a, **3a** bearing a weak electron-donating methoxyl group presents a broad absorption band centered at 451 nm in the low-energy region and 381/362/331/312 nm in the high energy region. In comparison, the other four compounds **3b**–**3e** display similar absorption profiles, while **3e** possesses a bathochromic shift absorption peak probably owing to the increase in the π-conjugation length with the introduction of cyano unit [32,33,34]. Compound **3a** exhibits a broad emission peak at 506 nm and the emission maxima and contours of the other four compounds **3b**–**3e** are almost the same (Figure 2b,d). The quantum yields are calculated to be 1.7% for **3a**, 1.9% for **3b**, 0.51% for **3c**, 0.27% for **3d**, 5.2% for **3e***,* respectively, by using 9,10-diphenylanthracene as a standard [28]. The fluorescence lifetimes (τ_s_) were recorded to be 8.40 ns for **3a**, 16.22 ns for **3b**, 31.32/5.06 ns for **3c**, 2.09/14.29 ns for **3d** and 12.50 ns for **3e**, respectively, by using a time-resolved fluorescence way (Appendix A). Clearly, molecules **3c** and **3d** display two lifetimes compared with the other three homologues. The low quantum yield of the **3d** containing iodine atom and the diexponential decay process of **3c** and **3d** should be ascribed to the heavy atom effect.

The electrochemical properties of the functionalized coronene derivatives were examined through cyclic voltammetry in anhydrous and degassed dichloromethane. As shown in Figure 2c, all of them exhibit one reversible oxidative wave with the potentials of 0.66 V for **3a**, 0.61 V for **3b**, 0.59 V for **3c**, 0.60 for **3d** and 0.72 V for **3e**, respectively, against ferrocene (Fc^+^/Fc), whereas no reduction waves could be monitored within the accessible scanning range in the dichloromethane. Accordingly, the HOMO energy levels are calculated to be −5.46 eV for **3a**, −5.41 eV for **3b**, −5.39 eV for **3c**, −5.40 eV for **3d**, −5.52 eV for **3e** on the basis of the first oxidation potentials. Molecular orbital calculations based on the B3lyp/def2SVP indicate that the HOMOs of all the compounds are spread over substituted tribenzo[*a*,*d*,*g*]coronene moiety and LUMOs are located on the substituted dibenzo[*fg*,*ij*]naphtho [1,2,3,4-*rst*]pentaphene unit (Figure 3 and Appendix A) [35,36,37,38]. Such observations suggest that the substituents do contribute to the orbitals to a lesser extent.

To examine the photoconductor properties, compounds **3b** and **3e** mixed with C_60_ were used as active layers to fabricate photodetector devices. As observed in Figure 4a,c, the blended systems of **3b**-C_60_ and **3e**-C_60_ were subjected to white light at varying illumination intensities, with the photocurrent increasing correspondingly. The maxima data of 0.031 µA for **3b**-C_60_ and 0.167 µA for **3e**-C_60_ at 200 mW/cm^2^ were generated when the mixture films were switched on and off. It should be stressed that no photocurrent was found that was white-light illumination-free. Such phenomena may be caused by the photo-induced charge transfer in the donor and acceptor systems. Meanwhile, film **3e**-C_60_ exhibited a higher photocurrent than film **3b**-C_60_, being close to the fluorescence spectra. In addition, the photoresponses to ON/OFF cycles were prompt, stable and reducible for both of them (Figure 4b,d). Such features of the tribenzocoronene derivatives endow an opportunity for them to be regarded as fascinating ingredients for a photo-controlled switch and photodetectors.

To expand the applications of such materials, the nonlinear optical properties of **3b** and **3e** were further studied through open aperture Z-scan technology under the Nd: YAG-based 532 nm wavelength nanosecond pulse laser irradiation [39]. Both of them present reverse saturation absorption (RSA) and the curves show symmetric peaks on both sides near the laser focus (Figure 5a,c). The nonlinear absorption coefficients (*β*_eff_) of **3b** and **3e** fluctuate with the energy density at the focal point, indicating that the ESA effect plays a major role in the RSA signal (Appendix A) [40]. The minimum normalized transmittance (*T*_min_) decreases gradually with the increase in incident energy. *T*_min_ of **3b** at 20.7 μJ, 40.6 μJ and 60.5 μJ are 90%, 78% and 67%, respectively. *T*_min_ of **3e** at 20.7 μJ, 40.6 μJ and 60.5 μJ are 76%, 67% and 59%, respectively. The onset optical limiting threshold (*F*_on_, the incident laser intensity when the normalized transmittance drops to 95%) of **3b** are 1.7 J cm^−2^, 0.845 J cm^−2^ and 0.137 J cm^−2^ under 20.7 μJ, 40.6 μJ and 60.5 μJ irradiation, respectively. The *F*_on_ of **3e** are 0.272 J cm^−2^, 0.146 J cm^−2^ and 0.215 J cm^−2^ under 20.7 μJ, 40.6 μJ and 60.5 μJ irradiation, respectively.

The possible mechanism of nonlinear optical processes is examined by measuring the nanosecond transient absorption spectroscopy. As shown in Figure 6, the timescale corresponding to the spectral change is approximately 50–1550 ns, which should be attributed to the effect of the triple excited state [41]. The peak position of the two compounds changed little with the delay time, indicating that the excited state absorption was generated in the same excited state, and no other processes occurred during the excited state absorption [42,43,44]. Both of them show similar spectral shapes, displaying wide excited state absorption bands after 440 nm and an isolated excited state absorption peak at 400 nm. There is also an isolated excited absorption peak of **3b** at 330 nm, but the excited absorption peak of **3e** is suppressed at this position, which scarcely shows a positive signal. The attenuation curves of **3b** and **3e** at the absorption peak of 480 nm are shown in Appendix A, and the attenuation lives of their triple excited states for **3d** and **3e** were 334.4 ns and 263.2 ns, respectively. On the whole, there was little difference between the two compounds, even though the excited state absorption peak of **3e** at 480 nm is slightly stronger than that of **3b**.

However, according to the Z-scan test results, **3e** has a lower *T*_min_ value than **3b**, which may be related to the different excitation pathways of the two compounds. Generally, molecular excitation is believed to result from the absorption of the band gap and the absorption of the defect level near the band gap for ns laser pulse irradiation. The UV-visible absorption spectra show that the ground state absorption (GSA) of **3b** and **3e** at the wavelength of 532 nm is relatively weak (Figure 2a), which is not conducive to the generation of excited molecules. In this case, the GSA of **3e** at 532 nm is slightly stronger than that of **3b**, which is beneficial to generating more excited molecules under laser irradiation, and may lead to a stronger RSA signal. In addition, the TPA excitation pathway of excited molecules cannot be excluded. The two-photon fluorescence (TPF) spectra at the excitation wavelength of 800 nm were tested and the logarithmic power-dependent TPF intensity curve displayed the linear-fitted slopes of 2.01 and 2.04 (Appendix A), which indicated that the TPF intensity exhibited a quadratic curve relationship with the excitation power, proving the existence of TPA [45]. Therefore, we reasonably speculate that the nonlinear absorption signals of **3b** and **3e** should be caused by TPA/GSA and ESA.

## 3. Materials and Methods

^1^H NMR and ^13^C NMR spectra were measured on a WNMR 400 spectrometer at 400 MHz for ^1^H and 100 MHz for ^13^C without any internal standard. The chemical shifts are labelled in ppm with δ of CDCl_3_ (7.26 ppm in ^1^H NMR and 77.16 in ^13^C NMR). MALDI-TOF mass spectra were performed on a Bruker Biflex III MALDI-TOF. UV-visible absorption and fluorescence spectra were carried out by using a 10 mm quartz cell on an Analytic Jena SPECORD 210 PLUS and Hitachi F-7000 spectrometers, respectively. Cyclic voltammetry investigations were performed on a CHI 630A electrochemical analyzer using a standard three-electrode cell containing a Pt working electrode, a Pt wire counter electrode and an Ag/AgNO_3_ reference electrode under a nitrogen atmosphere. Tetrabutylammonium hexafluorophosphate solution (0.1 M, anhydrous dichloromethane) was used as an electrolyte. The scan rate was 0.1 V s^−1^ and the redox potentials were labelled against the Fc^+^/Fc couple (a standard). The photoswitching behaviors were performed through an electrochemical workstation (Modulab XM, Solartron Analytical, UK) and the voltage was 0.5 V.

### 3.1. Synthesis of **3a**

TfOH (0.3 mL) was slowly dropped into a mixture of compound **2a** (30 mg, 0.04 mmol) and DDQ (39 mg, 0.17 mmol) in anhydrous dichloromethane (15 mL) at −30 °C under an argon atmosphere. After 7 min, methanol was added to quench the reaction. The mixture solution was partitioned between Na_2_CO_3_ solution/brine and methylene chloride. The organic layer was dried over Na_2_SO_4_ and evaporated in vacuo. The crude product was purified over silica gel column chromatography with petroleum ether (PE) as an eluent to produce a yellow solid (**3a**, 10 mg, 40%). ^1^H NMR (400 MHz, 298 K, CDCl_3_): δ = 9.41 (s, 1H), 8.77 (d, *J* = 7.6 Hz, 1H), 8.56 (s, 1H), 8.30 (d, *J* = 8.8 Hz, 1H), 8.21–8.17 (m, 2H), 8.11–8.04 (m, 3H), 7.96 (d, *J* = 7.2 Hz, 1H), 7.40–7.34 (m, 3H), 7.24 (d, *J* = 7.6 Hz, 2H), 4.06 (s, 3H), 1.75 (s, 9H), 1.59 (s, 9H). ^13^C NMR (100 MHz, 298 K, CDCl_3_): δ = 157.0, 147.4, 145.4, 140.5, 139.4, 132.5, 131.7, 131.3, 131.1, 130.0, 129.70, 129.67, 129.38, 129.35, 129.2, 128.8, 128.0, 127.96, 127.7, 127.1, 126.9, 126.0, 125.9, 124.6, 124.1, 124.0, 123.9, 123.6, 121.00, 120.98, 120.4, 117.1, 108.4, 55.9, 38.4, 35.8, 35.1, 31.9. HR–MS (MALDI–TOF): Calc. for C_45_H_36_O: [*m/z*] 592.2766, found: [*m/z*] 592.2756.

### 3.2. Synthesis of **2b**

A mixture of **1** (510 mg, 0.98 mmol), 2-amino-3-chlorobenzoic acid (204 mg, 1.19 mmol), isoamyl nitrate (0.2 mL) was stirred in anhydrous tetrachloroethane (TCE, 15 mL) at 150 °C under argon. After 24 h, the TCE was removed at a reduced pressure. The mixture was then partitioned between brine and methylene chloride. The organic layer was dried over Na_2_SO_4_ and evaporated in vacuo. The crude product was purified over silica gel column chromatography with PE as an eluent to give a light green solid (**2b**, 235 mg, 40%). ^1^H NMR (400 MHz, 298 K, CDCl_3_): δ = 7.94 (dd, *J* = 6.0 Hz, 2.0 Hz, 2H), 7.84 (s, 3H), 7.83 (d, *J* = 1.6 Hz, 1H), 7.74 (dd, *J* = 8.4 Hz, 1.2 Hz, 1H), 7.57–7.43 (m, 11H), 7.30 (q, *J* = 7.2 Hz, 1.2 Hz, 1H), 1.10 (s, 18 H). ^13^C NMR (100 MHz, 298 K, CDCl_3_): δ = 147.6, 147.1, 142.6, 142.1, 135.7, 135.0, 134.3, 133.1, 132.9, 131.8, 130.5, 130.4, 130.34, 130.27, 129.98, 129.5, 129.32, 129.26, 128.2, 127.9, 127.5, 127.2, 126.9, 126.2, 125.2, 124.4, 123.9, 122.7, 122.5, 34.9, 34.8, 31.5. HR–MS (MALDI–TOF): Calc. for C_44_H_37_Cl: [*m/z*] 600.2584, found: [*m/z*] 600.2574.

### 3.3. Synthesis of **3b**

TfOH (0.3 mL) was slowly dropped into a mixture of compound **2b** (20 mg, 0.03 mmol) and DDQ (22 mg, 0.1 mmol) in anhydrous dichloromethane (15 mL) at −30 °C under an argon atmosphere. After 5 min, methanol was added to quench the reaction. The mixture solution was partitioned between Na_2_CO_3_ solution and methylene chloride. The organic layer was dried over Na_2_SO_4_ and evaporated in vacuo. The crude product was purified over silica gel column chromatography with PE as an eluent to produce a yellow solid (**3b**, 13 mg, 66%). ^1^H NMR (400 MHz, 298 K, CDCl_3_): δ = 9.33 (d, *J* = 8.0 Hz, 1H), 8.96 (d, *J* = 1H), 8.86 (d, *J* = 7.6 Hz, 2H), 8.66 (d, *J* = 8.0 Hz, 1H), 8.57 (d, *J* = 8.4 Hz, 1H), 8.40 (q, *J* = 8.8 Hz, 2H), 8.32 (d, *J* = 8.0 Hz, 1H), 7.90 (d, *J* = 7.2 Hz, 1H), 7.83 (t, *J* = 7.6 Hz, 1H), 7.65 (t, *J* = 7.6 Hz, 1H), 7.58–7.51 (m, 2H), 7.44 (t, *J* = 7.2 Hz, 1H), 1.86 (s, 9H), 1.82 (s, 9H). ^13^C NMR (100 MHz, 298 K, CDCl_3_): δ = 146.0, 145.6, 132.9, 132.7, 132.0, 131.8, 130.8, 130.7, 129.8, 129.6, 129.3, 129.14, 129.07, 129.0, 128.8, 128.2, 127.7, 127.4, 127.1, 126.7, 126.6, 126.3, 125.9, 125.5, 124.8, 123.5, 123.2, 122.8, 122.7, 122.5, 120.8, 38.7, 38.6, 35.2, 35.1. HR–MS (MALDI–TOF): Calc. for C_44_H_33_Cl: [*m/z*] 596.2271, found: [*m/z*] 596.2263.

### 3.4. Synthesis of **2d**

A mixture of **1** (1.5 g, 2.89 mmol), 2-amino-3-iodobenzoic acid (913 mg, 3.47 mmol), isoamyl nitrate (1.0 mL) was stirred in anhydrous tetrachloroethane (TCE, 15 mL) at 150 °C under argon. After 24 h, TCE was removed at a reduced pressure. The mixture was then partitioned between brine and methylene chloride. The organic layer was dried over Na_2_SO_4_ and evaporated in vacuo. The crude product was purified over silica gel column chromatography with PE as an eluent to produce a light green solid (**2d**, 1.13 g, 56%). ^1^H NMR (400 MHz, 298 K, CDCl_3_): δ = 8.19 (dd, *J* = 7.2 Hz, 1.2 Hz, 1H), 7.91 (d, *J* = 2.0 Hz, 1H), 7.84 (s, 4H), 7.83 (d, *J* = 2.0 Hz, 1H), 7.77 (dd, *J* = 8.8 Hz, 1.2 Hz, 1H), 7.56–7.43 (m, 10H), 7.01 (q, *J* = 7.2 Hz, 1H), 1.11 (s, 9H), 1.10 (s, 9H). ^13^C NMR (100 MHz, 298 K, CDCl_3_): δ = 147.6, 147.0, 141.8, 141.6, 140.5, 136.5, 135.3, 135.2, 134.1, 133.1, 132.6, 130.4, 130.0, 129.8, 129.3, 128.6, 128.0, 127.9, 127.5, 127.4, 127.1, 126.9, 125.9, 124.4, 123.9, 122.7, 122.5, 91.9, 34.9, 34.8, 31.5. HR–MS (MALDI–TOF): Calc. for C_44_H_37_I: [*m/z*] 692.1940, found: [*m/z*] 692.1934.

### 3.5. Synthesis of **3d**

TfOH (0.3 mL) was slowly dropped into a mixture of compound **2b** (20 mg, 0.03 mmol) and DDQ (20 mg, 0.09mmol) in anhydrous dichloromethane (15 mL) at −30 °C under an argon atmosphere. After 5 min, methanol was added to quench the reaction. The mixture solution was partitioned between Na_2_CO_3_ solution/brine and methylene chloride. The organic layer was dried over Na_2_SO_4_ and evaporated in vacuo. The crude product was purified over silica gel column chromatography with PE as an eluent to produce a light yellow solid (**3d**, 18 mg, 90%). ^1^H NMR (400 MHz, 298 K, CDCl_3_): δ = 9.43 (d, *J* = 8.0 Hz, 1H), 8.96 (d, *J* = 8.4 Hz, 1H), 8.86 (d, *J* = 6.8 Hz, 2H), 8.67 (d, *J* = 8.0 Hz, 1H), 8.58 (d, *J* = 8.0 Hz, 1H), 8.46–8.37 (m, 4H), 7.66 (t, *J* = 7.6 Hz, 1H), 7.60–7.52 (m, 3H), 7.48 (d, *J* = 8.0 Hz, 1H), 1.86 (s, 9H), 1.82 (s, 9H). ^13^C NMR (100 MHz, 298 K, CDCl_3_): δ = 146.1, 145.7, 140.7, 132.7, 132.1, 132.0, 130.4, 130.3, 130.0, 129.22, 129.20, 129.1, 129.0, 128.8, 127.2, 127.1, 126.7, 126.3, 125.0, 123.5, 123.2, 123.1, 122.7, 122.6, 120.83, 120.79, 97.5, 38.7, 38.6, 35.2, 35.1. HR–MS (MALDI–TOF): Calc. for C_44_H_33_I: [*m/z*] 688.1627, found: [*m/z*] 688.1619.

### 3.6. Synthesis of **3e**

A mixture of **3c** (100 mg, 0.16 mmol) and CuCN (28 mg, 0.31 mmol) was stirred in anhydrous NMP (6 mL) at 180 °C under argon. After 3 d, ammonium ferrous sulfate solution was added when the mixture solution was cooled to 60 °C for 2 h. The solution was then cooled down to room temperature and was partitioned between brine and methylene chloride. The organic layer was dried over Na_2_SO_4_ and evaporated in vacuo. The crude product was purified over silica gel column chromatography with PE and dichloromethane (*v/v*, 8:1) as an eluent to produce a light yellow solid (**3e**, 22 mg, 24%). ^1^H NMR (400 MHz, 298 K, CDCl_3_): δ = 9.61 (d, *J* = 8.0 Hz, 1H), 8.91–8.87 (m, 3H), 8.70 (d, *J* = 8.0 Hz, 1H), 8.63 (dd, *J* = 10.8 Hz, 8.4 Hz, 2H), 8.45 (dd, *J* = 11.2 Hz, 8.4 Hz, 2H), 8.27 (d, *J* = 6.8 Hz, 1H), 7.95 (t, *J* = 8.0 Hz, 1H), 7.68 (t, *J* = 8.0 Hz, 2H), 7.56 (q, *J* = 8.0 Hz, 2H), 1.85 (s, 9H), 1.82 (s, 9H). ^13^C NMR (100 MHz, 298 K, CDCl_3_): δ = 146.3, 145.8, 134.1, 133.9, 132.9, 132.24, 132.03, 130.7, 129.8, 129.7, 129.23, 129.17, 128.94, 128.93, 128.6, 128.54, 128.47, 127.6, 127.4, 127.2, 126.9, 126.6, 125.7, 125.6, 125.1, 124.7, 123.7, 123.5, 123.3, 123.2, 122.9, 122.7, 120.9, 120.8, 120.3, 111.4, 38.8, 35.2, 35.1. HR–MS (MALDI–TOF): Calc. for C_45_H_33_N: [*m/z*] 587.2613, found: [*m/z*] 587.2602.

## 4. Conclusions

In summary, we have designed and synthesized five novel coronene-containing π-systems bearing different substituents. Such an investigation highlights a significant effect of the substituents on the absorption, emission and redox properties of **3a**–**3e**. Molecule **3d** has the lowest quantum yield owing to the strong heavy atom effect of iodine. The photocurrent response of **3e**-C_60_ is superior to that of **3b**-C_60_, which is assigned to the higher quantum yield of **3e**, leading to a highly efficient photo-induced charge transfer in the donor and acceptor system. The expanded applications suggest that the synthesized compounds **3b** and **3e** have a positive optical limiting performance resulting from GSA and ESA phenomena. Further examination of the post-functionalization of such key building blocks for approaching large curved PAHs with attractive optoelectronic properties are currently being undertaken in our laboratory.

## Data Availability

The data presented in this study are available in the paper. The CCDC 2232268 contains supplementary crystallographic data for this paper. These data can be obtained free of charge via http://www.ccdc.cam.ac.uk/conts/retrieving.html (or from the Cambridge Crystallographic Data Centre, 12, Union Road, Cambridge CB2 1EZ, UK; fax: +44 1223 336033).

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
