# Peer review of "Synthesis, Photoswitching Behavior and Nonlinear Optical Properties of Substituted Tribenzo[*a*,*d*,*g*]coronene"

_molecules, 2023, doi:10.3390/molecules28031419_

Round 1

Reviewer 1 Report

The article presents interesting material. However it is not presented clearly enough.  

1/There  is no clear correlation of properties with the structure of compounds

2/ It is also desirable to explane how the conclesion about  nolinear properties from two-photon absorbtion is made

3. . Tye position of materials and methods section (4) after conclusions section(3) loors very strange

Author Response

Reviewer 1 Comments and Suggestions for Authors

The article presents interesting material. However it is not presented clearly enough.

Response: We thank the reviewer for his positive comments and strong support.

Comment (1): There is no clear correlation of properties with the structure of compounds.

Response: Thanks for the reviewer’s comments. Structure-activity relationship analysis shows that optical limiting performance is the comprehensive reflection of GSA and ESA phenomenon. The two molecules have similar ESA properties, therefore, we believe that the regulation of nonlinear absorption properties is caused by different substituents changing the GSA properties of the two molecules.

Based on the analysis, the conclusion was described as follows.

In summary, we have designed and synthesized five novel coronene-containing π-systems bearing different substituents. Such investigation highlight a significant effect of the substituents on the absorption, emission and redox properties of 3a-3e. Molecule 3d has a lowest quantum yield owing to the strong heavy atom effect of iodine. The photocurrent response of 3e-C60 is superior to that of 3b-C60, which is assigned to the higher quantum yield of 3e, leading to highly efficient photo-induced charge transfer in the donor and acceptor system. The expanded applications suggest that the synthesized compounds 3b and 3e have positive optical limiting performance resulting from GSA and ESA phenomena. Further examination of post-functionalization of such key building blocks for approaching large curved PAHs with attractive optoelectronic property are currently being undertaken in our laboratory.

Comment (2): It is also desirable to explain how the conclusion about nonlinear properties from two-photon absorption is made.

Response: Thanks for the reviewer’s comments. Nanosecond pulsed lasers have a much lower peak power density than femtosecond pulsed lasers and thus ESA are more likely to occur than TPA. Therefore, the mainstream explanation of nanosecond pulse laser optical liming signal is GSA+ESA or TPA+ESA, rather than a separate TPA process. In this paper, the two-photon fluorescence spectra of two kinds of molecules were measured by 800 nm femtosecond laser excitation, and the existence of two-photon absorption was proved. Therefore, it is reasonable to speculate that GSA+ESA and TPA+ESA are both sources of nonlinear absorption signals.

Comment (3): Type position of materials and methods section (4) after conclusions section (3) loors very strange.

Response: Thanks for the reviewer’s comments. According to the suggestion and guideline of such journal, ‘(4) conclusion’ was placed after ‘(3) Materials and Methods’ and please see the revised manuscript.

Reviewer 2 Report

Dear Editor

This work is highly distinguished and merits publication after several minor revisions, obligatory and optional, which are noted below:

Obligatory revisions:

Comment 1: 

All aspects of the manuscript should be revised for correcting linguistic errors. The manuscript should be fine-tuned by the authors.

Comment 2:

The (Materials and Methods) section should be placed after the introduction.

Comment 3:

In lines (167-169), the authors stated:

The peak position of the two compounds changed little with the delay time, indicating that the excited state of absorption was generated in the same excited state, and no other processes occurred during the excited state of absorption.

Please support this statement with other interpretations in the literature. Clearly argued?

Comment 4:

Please amend the error on line 151.

Comment 5:

Please clarify the figures as much as possible. Please choose colors that appear clearly in the print.

Optional revisions:

Comment 1: 

The title is not suggestive. Is it possible to change the title to an attractive one that indicates the importance of the content of the manuscript?

Comment 2:

Can authors include specifications for the Gaussian program used in the simulation?

End,,,

Author Response

Reviewer 2 This work is highly distinguished and merits publication after several minor revisions, obligatory and optional, which are noted below:

Response: We thank the reviewer for his positive comments and strong support.

Obligatory revisions:

Comment (1): All aspects of the manuscript should be revised for correcting linguistic errors. The manuscript should be fine-tuned by the authors.

Response: Thanks for the reviewer’s comments. Following the suggestion, we detailly revised this manuscript and please see the ‘track change’ file.

Comment (2): The (Materials and Methods) section should be placed after the introduction.

Response: Thanks for the reviewer’s comments. According to the suggestion and guideline of such journal, ‘(4) conclusion’ was placed after ‘(3) Materials and Methods’ and please see the revised manuscript.

Comment (3): In lines (167-169), the authors stated: The peak position of the two compounds changed little with the delay time, indicating that the excited state of absorption was generated in the same excited state, and no other processes occurred during the excited state of absorption. Please support this statement with other interpretations in the literature. Clearly argued?

Response: Thanks for the reviewer’s comments. Transient absorption (TA) spectroscopy is a powerful technique to provide further information on the triplet excited-state absorption and decay characteristics, especially to figure out the spectral regions where the excited-state absorption is stronger than that of the ground state. The TA spectral shapes of the two compounds in Figure 6 are similar, which is related to the similar structure of the two compounds. The peak shape of TA spectrum did not change with the delay time. These facts suggest that the TA of 3b and 3e likely arises from the same excited state [42-44].

https://doi.org/10.1021/jp506765k

https://doi.org/10.1021/jp411259f

https://doi.org/10.1021/acs.inorgchem.0c00961

Comment (4): Please amend the error on line 151.

Response: Thanks for the reviewer’s comments. According to the suggestion, ‘studies’ change into ‘studied’.

Comment (5): Please clarify the figures as much as possible. Please choose colors that appear clearly in the print.

Response: Thanks for the reviewer’s comments. Following the suggestion, we re-drew Figure 2, 4 and 6 and please see the revised manuscript (below).

Figure 2. (a) UV-vis absorption and (b) fluorescence spectra of 3a, 3b, 3c, 3d, 3e in dichloromethane. (c) Cyclic voltammetry of 3a-3e in dry dichloromethane with tetrabutylammonium hexafluorophosphate as a supporting electrolyte. (d) Photographs with illumination at 365 nm.

Figure 4. Dependence of current on different input light intensities (a) 3b-C60, (c) 3e-C60 and time dependence of dynamic photoresponse properties of films (b) 3b-C60, (d) 3e-C60 upon the irradiation of 200 mW/cm2 white light.

Figure 6. Time-resolved transient difference absorption of 3b (a) and 3e (b). The pump wavelength is 355 nm.

Optional revisions:

Comment 1: The title is not suggestive. Is it possible to change the title to an attractive one that indicates the importance of the content of the manuscript?

Response: Thanks for the reviewer’s comments. According to the reviewer’s suggestion, the title was described as follows. ‘Synthesis, Photoswitching Behavior and Nonlinear Optical Properties of Substituted Tribenzo[a,d,g]coronene’.

Comment 2: Can authors include specifications for the Gaussian program used in the simulation?

Response: Thanks for the reviewer’s comments. The density functional theory calculation was further carried out at the B3lyp/def2SVP level to realize the electronic structures of the as-prepared compounds. The description is presented as follows.

Molecular orbital calculations based on the B3lyp/def2SVP indicate that the HOMOs of all the compounds are spread over substituted tribenzo[a,d,g]coronene moiety and LUMOs are located on substituted dibenzo[fg,ij]naphtho[1,2,3,4-rst]pentaphene unit (Figure 3) [35].

(32) Frisch, M. J.; Trucks, G. W.; Schlegel, H. B.; Scuseria, G. E.; Robb, M. A.; Cheeseman, J. R.; Scalmani, G.; Barone, V.; Mennucci, B.; Petersson, G. A.; Nakatsuji, H.; Caricato, M.; Li, X.; Hratchian, H. P.; Izmaylov, A. F.; Bloino, J.; Zheng, G.; Sonnenberg, J. L.; Hada, M.; Ehara, M.; Toyota, K.; Fukuda, R.; Hasegawa, J.; Ishida, M.; Nakajima, T.; Honda, Y.; Kitao, O.; Nakai, H.; Vreven, T.; Montgomery Jr, J. A.; Peralta, J. E.; Ogliaro, F.; Bearpark, M.; Heyd, J. J.; Brothers, E.; Kudin, K. N.; Staroverov, V. N.; Keith, T.; Kobayashi, R.; Normand, J.; Raghavachari, K.; Rendell, A.; Burant, J. C.; Iyengar, S. S.; Tomasi, J.; Cossi, M.; Rega, N.; Millam, J. M.; Klene, M.; Knox, J. E.; Cross, J. B.; Bakken, V.; Adamo, C.; Jaramillo, J.; Gomperts, R.; Stratmann, R. E.; Yazyev, O.; Austin, A. J.; Cammi, R.; Pomelli, C.; Ochterski, J. W.; Martin, R. L.; Morokuma, K.; Zakrzewski, V. G.; Voth, G. A.; Salvador, P.; Dannenberg, J. J.; Dapprich, S.; Daniels, A. D.; Farkas, O.; Foresman, J. B.; Ortiz, J. V.; Cioslowski, J.; Fox, D. J. Gaussian, Inc., Wallingford CT, Gaussian 09, revision B.01. 2010.

Reviewer 3 Report

Xueqing Li et.al reported that a family of expanded coronene derivatives bearing various substituents (3) was constructed through the Diels-Alder reaction, followed by the Scholl oxidation, where the molecular structure of 3b was determined via single crystal X-ray diffraction analysis. The effect of substitution on optical and electrochemical property was systematically investigated, with the assistance of theoretical calculations. Moreover, the thin films of the resulting molecules 3b and 3e complexed with fullerene produced strong photocurrent response upon irradiation of white light. In addition, 3b and 3e exhibit positive nonlinear optical response resulting from the two-photon absorption and excited state absorption processes.

Recommendation: In this writing, many grammatical and scientific mistakes are observed. So, I recommend this manuscript for publishing in Molecules with following suggestions:

1.     Author must maintain line spacing of 1.5 throughout the manuscript.

2.     The whole manuscript is written in Palatino Linotype writing style but the size of the writing is not same throughout the manuscript, abstract is written in size 9 while remaining article is written with the size 10. So, author must maintain the same size of writing throughout the manuscript.

3.     It is observed that, some sentences in the manuscript are too long. Long and convoluted sentences are harder to understand and affect the readability and comprehension. Use moderate size sentences uniformly throughout the manuscript.

4.     Italicize the units and the words like “via, etc, i.e., vs.” throughout the manuscript and centralize the equations and figures as well.

5.     In figure 5 units are not italicized and the figures 4 and 5 are also distorted and not clear. Please add the figures again with correction.

6.     In introduction part, only abbreviations are used i.e TCNQ, F4TCNQ. Full form of these abbreviations must be mentioned in the manuscript.

7.     At the start of second paragraph of introduction the word “widely” is written, which is not suitable so replace it with “wide” or another suitable word.

8.     In Scheme 1 the description is given like “Synthetic route to 3” which is not suitable. Please write it in another suitable way by mentioning “3c and 3e” in it.

9.     In the paragraph below the figure 1, the word “imply” is used in the sentence “Molecule 3b can stack in column style, where the distance between the naphthalene in the pyrene unit is 3.63 Å (Figure 1c), imply that π-π stacking interaction is absent.” Author must write it as “which implies that”.

10.  In the manuscript the words like “the as-synthesized” and “the as-prepared” are written in different paragraphs which are not suitable. So please write them as “the synthesized” or “as-synthesized”.

11.  In conclusion, the word “studied” is used in the following sentence “The studied on 3a-3e highlight a significant effect of the substituents on the absorption, emission and redox properties” which is not suitable please replace it with another suitable word according to the sentence.

12.  In the description of synthesis of 3b, the words “mixture” and “solution are used at a time in the sentence “The mixture solution was partitioned between Na2CO3 solution/brine and methylene chloride” which is not suitable. So, author should use any one of them.

13.  In the manuscript the terms “UV/vis” and “UV-vis” are used. So please mention it in a uniform manner throughout the manuscript i.e UV-visible.

14.  Cite following DOI numbers in Frontier Molecular Orbitals (FMOs) in order to explain the energy gap between the molecular orbitals;

https://doi.org/10.5012/bkcs.2014.35.5.1391

https://doi.org/10.1016/j.arabjc.2014.11.007

https://doi.org/10.1016/j.molstruc.2021.130650

15.  Moreover, cite the following articles in the portion of electronic transitions for the illustration of absorption maxima, which are more relevant to your UV-Vis studies

https://doi.org/10.1002/poc.3427

https://doi.org/10.1016/j.jscs.2014.06.001

https://doi.org/10.1016/j.arabjc.2021.103295

Author Response

Reviewer 3 Xueqing Li et.al reported that a family of expanded coronene derivatives bearing various substituents (3) was constructed through the Diels-Alder reaction, followed by the Scholl oxidation, where the molecular structure of 3b was determined via single crystal X-ray diffraction analysis. The effect of substitution on optical and electrochemical property was systematically investigated, with the assistance of theoretical calculations. Moreover, the thin films of the resulting molecules 3b and 3e complexed with fullerene produced strong photocurrent response upon irradiation of white light. In addition, 3b and 3e exhibit positive nonlinear optical response resulting from the two-photon absorption and excited state absorption processes.

Response: We thank the reviewer for his positive comments and strong support.

Recommendation: In this writing, many grammatical and scientific mistakes are observed. So, I recommend this manuscript for publishing in Molecules with following suggestions:

Comment (1): Author must maintain line spacing of 1.5 throughout the manuscript.

Response: Thanks for the reviewer’s comments. According to the suggestion, we maintain the line spacing of 1.5 in the revised manuscript.

Comment (2): The whole manuscript is written in Palatino Linotype writing style but the size of the writing is not same throughout the manuscript, abstract is written in size 9 while remaining article is written with the size 10. So, author must maintain the same size of writing throughout the manuscript.

Response: Thanks for the reviewer’s comments. According to the suggestion, we change the font size of ABSTRACT and ADDRESS parts to size 10.

Comment (3): It is observed that, some sentences in the manuscript are too long. Long and convoluted sentences are harder to understand and affect the readability and comprehension. Use moderate size sentences uniformly throughout the manuscript.

Response: Thanks for the reviewer’s comments. According to the reviewer’s suggestion, we have revised these sentences and please see highlight in the revised manuscript.

Comment (4): Italicize the units and the words like “via, etc, i.e., vs.” throughout the manuscript and centralize the equations and figures as well.

Response: Thanks for the reviewer’s comments. They have been revised and please see the revised manuscript.

Comment (5): In figure 5 units are not italicized and the figures 4 and 5 are also distorted and not clear. Please add the figures again with correction.

Response: Thanks for the reviewer’s comments. According to the reviewer’s suggestion, they have been revised and please see the revised manuscript.

Comment (6): In introduction part, only abbreviations are used i.e TCNQ, F4TCNQ. Full form of these abbreviations must be mentioned in the manuscript.

Response: Thanks for the reviewer’s comments. According the reviewer’s suggestion, the full name of 7,7,8,8-tetracyanoquinodimethane (TCNQ), 2,3,5,6-tetrafluoro-7,7,8,8-tetracyanoquinodimethane (F4TCNQ) were added to the manuscript.

Comment (7): At the start of second paragraph of introduction the word “widely” is written, which is not suitable so replace it with “wide” or another suitable word.

Response: Thanks for the reviewer’s comments. According to the suggestion, ‘widely’ was replaced with ‘wide’.

Comment (8): In Scheme 1 the description is given like “Synthetic route to 3” which is not suitable. Please write it in another suitable way by mentioning “3c and 3e” in it.

Response: Thanks for the reviewer’s comments. According to the suggestion, ‘Scheme 1. Synthetic route to 3.’ was changed into ‘Scheme 1. Synthetic route to 3a, 3b, 3c, 3d and 3e.’.

Comment (9): In the paragraph below the figure 1, the word “imply” is used in the sentence “Molecule 3b can stack in column style, where the distance between the naphthalene in the pyrene unit is 3.63 Å (Figure 1c), imply that π-π stacking interaction is absent.” Author must write it as “which implies that”.

Response: Thanks for the reviewer’s comments. It has been revised.

Comment (10): In the manuscript the words like “the as-synthesized” and “the as-prepared” are written in different paragraphs which are not suitable. So please write them as “the synthesized” or “as-synthesized”.

Response: Thanks for the reviewer’s comments. They have been revised.

Comment (11): In conclusion, the word “studied” is used in the following sentence “The studied on 3a-3e highlight a significant effect of the substituents on the absorption, emission and redox properties” which is not suitable please replace it with another suitable word according to the sentence.

Response: Thanks for the reviewer’s comments. The sentence was revised as follows in the revised manuscript. ‘Such investigation highlight a significant effect of the substituents on the absorption, emission and redox properties of 3a-3e.’

Comment (12): In the description of synthesis of 3b, the words “mixture” and “solution are used at a time in the sentence “The mixture solution was partitioned between Na2CO3 solution/brine and methylene chloride” which is not suitable. So, author should use any one of them.

Response: Thanks for the reviewer’s comments. According to the suggestion, ‘/brine’ was deleted in the revised manuscript.

Comment (13): In the manuscript the terms “UV/vis” and “UV-vis” are used. So please mention it in a uniform manner throughout the manuscript i.e UV-visible.

Response: Thanks for the reviewer’s comments. According to the suggestion, ‘UV/vis’ and ‘UV-vis’ were changed into ‘UV-visible’.

Comment (14): Cite following DOI numbers in Frontier Molecular Orbitals (FMOs) in order to explain the energy gap between the molecular orbitals;

https://doi.org/10.5012/bkcs.2014.35.5.1391

https://doi.org/10.1016/j.arabjc.2014.11.007

https://doi.org/10.1016/j.molstruc.2021.130650

Response: Thanks for the reviewer’s comments. According to the suggestion, such interesting papers have been added to the revised manuscript as reference 36-38.

Comment (15): Moreover, cite the following articles in the portion of electronic transitions for the illustration of absorption maxima, which are more relevant to your UV-Vis studies

https://doi.org/10.1002/poc.3427

https://doi.org/10.1016/j.jscs.2014.06.001

https://doi.org/10.1016/j.arabjc.2021.103295

Response: Thanks for the reviewer’s comments. According to the suggestion, such interesting papers have been added to the revised manuscript as reference 32-34. By the way, the other references were re-arranged in order.

Round 2

Reviewer 3 Report

Article is improved and it may be publishable